# Perceived Risk of Binge Drinking among Older Alcohol Users: Associations with Alcohol Use Frequency, Binge Drinking, Alcohol Use Disorder, and Alcohol Treatment Use

**DOI:** 10.3390/ijerph21081081

**Published:** 2024-08-16

**Authors:** Namkee G. Choi, C. Nathan Marti, Bryan Y. Choi

**Affiliations:** 1Steve Hicks School of Social Work, University of Texas at Austin, 1925 San Jacinto Blvd, Austin, TX 78712, USA; nate.marti@utexas.edu; 2Department of Emergency Medicine, Philadelphia College of Osteopathic Medicine and BayHealth, Dover, DE 19901, USA; bryan.choi@bayhealth.org

**Keywords:** binge drinking, alcohol risk perception, alcohol use disorder, alcohol treatment

## Abstract

Despite the high prevalence of alcohol use and binge drinking among older adults, little research has been conducted on the association between their alcohol risk perception and alcohol use patterns. Using data on past-year alcohol users aged 50 and older (N = 6693) in the 2022 National Survey on Drug Use and Health, we examined the (1) associations between risk perception of binge alcohol use 1–2 times a week and alcohol use frequency, binge use frequency, and alcohol use disorder (AUD), and (2) the association between alcohol treatment use and risk perception. About 40% of past-year alcohol users perceived great risk of binge alcohol use 1–2 times a week, and 27% of past-year users had binge drinking in the past month. Multivariable analyses showed the negative association between great risk perception and alcohol use frequency (IRR = 0.60, 95%CI = 0.48–0.74 for daily use) and past-month binge alcohol use (IRR = 0.33, 95%CI = 0.19–0.57 for 6–19 days of binge use). The odds of great risk perception were also lower among those with mild AUD. Risk perception was not significantly associated with alcohol treatment. The lower likelihood of risk perception among problematic alcohol users and low treatment use is concerning. Education and interventions to reduce harm from alcohol are needed.

## 1. Introduction

Alcohol has been ranked as the most harmful of all drugs to those who use it and to society through related violence, accidents, and crimes [1,2,3,4]. Negative health effects of alcohol on users tend to become more pronounced in later life because liver enzymes that metabolize alcohol and other drugs become less efficient, and the central nervous system becomes more sensitive to drugs [5]. In addition, age-related decreases in body water and increased fat contribute to higher blood alcohol concentrations (BAC) in older drinkers, resulting in higher sensitivity to acute alcohol-induced impairments in memory, coordination, reaction time, and driving performance [6,7]. Alcohol-related problems among older adults include not only medical emergencies from acute intoxication and related injuries (e.g., falls) and dangerous interactions with medications, but also liver diseases, cardiomyopathy and other cardiovascular diseases, sleep disturbances, cancer, cognitive decline, and increased mortality from long-term alcohol use [7,8,9,10,11,12].

Despite its harmful effects on the aging body and mind, epidemiologic data show that over 60% (61.4% in 2021) of those age 50 and older report alcohol use in the preceding 12 months, close to 50% (47.1% in 2021) used alcohol in the past month, and about 17% (16.4% in 2021) had binge drinking (having 5+ drinks for men and 4+ drinks for women within 1–2 h) in the past month [13]. Our own analysis of the 2021 National Survey on Drug Use and Health (NSDUH) data also found that 7.7% of those age 50 and older met the *Diagnostic and Statistical Manual of Mental Disorders, 5th edition* (*DSM-5*) [14] alcohol use disorder (AUD) criteria, and only 4.1% of those with AUD received any behavioral or pharmacological alcohol treatment. The low rate of alcohol treatment use indicates a significant treatment gap [15]. Moreover, the actual rate of AUD in older adults is estimated to be higher given that some *DSM-5* criteria, such as effects on occupational roles, are not applicable to older adults who are not in the labor force [16].

Although the true rate of AUD in older adults is not known, research has shown that binge drinking among older men (age 65+) increased between 2015 and 2019 [17]. While the overall increases in alcohol consumption were sharper among older women [18], their binge drinking rate remained stable [17]. The binge drinking rate in the 65+ age group was also higher among non-Hispanic African Americans than whites, tobacco users, cannabis users, and those who had visited the emergency department in the past year [19]. Another study found that between 2015 and 2019, combined binge drinking and cannabis use increased in the 50+ age group, with the sharpest increase in the 65+ age group [20].

Research shows that older-adult problematic drinkers may not recognize the risks associated with alcohol, as they tend to have the self-image of a controlled and responsible drinker and appreciate alcohol use in sustaining social and leisure activities that are important to health and well-being in later life [21,22,23]. Engagement in social activities involving alcohol was associated with late-onset AUD for older adults who experienced a marked loss of identity following retirement [24]. Low alcohol-related risk perception may in part be responsible for the continued harmful use of alcohol among older adults with chronic health conditions [25], driving under the influence (DUI) of alcohol [26], and the low rate of alcohol treatment use [27]. Fairman and Early’s study [27] based on 2015–2018 NSDUH data also found that binge drinking was associated with lower treatment use in the 50+ age group.

In general, the literature on older alcohol users’ risk perception is scant. We only found one study that included older adults in its examination of changes in alcohol and cannabis risk perceptions from 2002 to 2019. The study showed that unlike cannabis-related risk perceptions (i.e., smoking cannabis 1–2 times a week), which decreased across all ages over time, alcohol-related risk perceptions (having 5+ drinks 1–2 times a week) stayed relatively stable during the same period with only negligible increases without any age group differences [28]. Given the high prevalence of binge drinking and AUD in the 50+ age group, it is important to examine older alcohol users’ risk perception related to binge drinking and its associations with alcohol use frequency, binge drinking frequency, AUD, and alcohol treatment use.

In the present study, based on the 2022 NSDUH, we first examined the prevalence of past-year alcohol use and use frequency, binge use in the past month, past-year AUD, and risk perceptions regarding binge alcohol use 1–2 times a week in the 50+ age group in general, and then among past-year alcohol users. We also examined potential differences in prevalence between the 50–64 and the 65+ age groups. Secondly, focusing on past-year alcohol users, we provided descriptive statistics comparing those who perceived a great risk of binge use and those who did not. Thirdly, we examined the associations of risk perception with alcohol use frequency, binge use frequency, and AUD. Fourthly, we examined the association between past-year alcohol treatment use and risk perception, with binge alcohol use frequency, AUD, DUI, physical and mental health status, other substance misuse, and sociodemographic factors as covariates. These findings add to our knowledge base regarding alcohol-related risk perception among older-adult alcohol users.

## 2. Materials and Methods

### 2.1. Data Source

The NSDUH is an annual epidemiologic survey, funded by the U.S. Substance Abuse and Mental Health Services Administration, of the civilian, non-institutionalized, age 12+ population to measure the prevalence of substance use, mental and substance use disorders, behavioral health treatment, and physical/functional health and healthcare use. Because of the ongoing COVID-19 pandemic, data collection in 2022 was carried out using both in-person and web-based modes. However, from 3 February 2022, in-person data collection became available in all states and counties, and in-person data collection commenced after potential respondents were first given the opportunity to complete the survey via the web [29]. The overall percentage of interviews that were completed via the web was 40.7% for 2022. The 2022 NSDUH public use data set includes responses from a total of 59,069 individuals who completed an in-person or web-based NSDUH survey. In the present study, we included 10,560 respondents aged 50 and older and then focused on 6693 past-year alcohol users. Analyses of these de-identified public-use data were exempt from the authors’ institutional review board review.

### 2.2. Measures

*Alcohol risk perception*: All NSDUH respondents were asked a question, “How much do people risk harming themselves physically and in other ways when they have five or more drinks of an alcoholic beverage once or twice a week?” The response categories were no risk, slight risk, moderate risk, and great risk. In this study, we used a dichotomous category of great risk versus no/slight/moderate risk.

*Past-year alcohol use, use frequency, and binge use*: In the NSDUH, alcohol use was first assessed with a question, “Have you ever, even once, had a drink of any type of alcoholic beverage? Please do not include times when you only had a sip or two from a drink,” with a ‘drink’ referring to a can or bottle of beer, a glass of wine or a wine cooler, a shot of liquor, or a mixed drink with liquor in it. Those who answered this question affirmatively were queried about first use, time since last use, the number of days that they used alcohol in the past month and year. *Past-year use* refers to use within the past 12 months, and *use frequency* among past-year users is categorized as 1–11 days, 12–49 days, 50–100 days, 100–299 days, and 300–365 days. *Past-month binge use* is defined as consuming 4 or more (for females) or 5 or more (for males) drinks on the same occasion. *Binge use frequency* is categorized as 1–2 days, 3–5 days, 6–19 days, and 20–30 days. We reported past-month heavy drinking (5+ drinks on any day or 15+ drinks per week for males and 4+ drinks on any day and 8+ drinks per week) for descriptive purposes only.

*Past-year alcohol use disorder (AUD)*: The NSDUH substance use disorders are based on the *DSM-5* criteria [14]. The 11 criteria for past-year AUD are as follows: (a) often used in larger amounts or over a longer period than was intended; (b) a persistent desire or unsuccessful efforts to cut down or control alcohol use; (c) spending a great deal of time on activities necessary to obtain alcohol, use alcohol, or recover from its effects; (d) craving, or a strong desire or urge to use alcohol; (e) recurrent use resulting in a failure to fulfill obligations at work, school, or home; (f) continued use despite having persistent or recurrent social or interpersonal problems caused or exacerbated by the effects of alcohol; (g) important social, occupational, or recreational activities given up or reduced because of alcohol use; (h) recurrent use in situations in which it was physically hazardous; (i) continued use despite knowledge of having a persistent or recurrent physical or psychological problem that was likely to have been caused or exacerbated by alcohol; (j) indicators of tolerance including a need for markedly increased amounts of alcohol to achieve intoxication or the desired effect, or a markedly diminished effect with continued use of the same amount; and (k) indicators of withdrawal, including sweating /rapid heartbeat, trembling hands, sleep disturbances, vomiting/upset stomach, audiovisual or other sensory hallucinations, inability to sit still, feeling anxious; or used alcohol or a closely related substance, such as a benzodiazepine, to avoid or relieve withdrawal symptoms). Any AUD was diagnosed when the respondent met 2 of these 11 criteria, with severity levels indicated as mild (meeting 2–3 criteria), moderate (meeting 4–5 criteria), or severe (meeting 6–11 criteria).

*Physical and mental health conditions and other substance use problems*: These were measured using the following criteria: (a) the number (0–10) of diagnosed chronic illnesses included asthma, cancer, chronic obstructive pulmonary disease, diabetes, heart disease, hepatitis, HIV/AIDS, hypertension, kidney disease, and liver disease; (b) any emergency department (ED) visit last year; (c) any past-year mental illness categorized as none, mild, moderate, serious (any mental, behavioral, or emotional disorder—excluding developmental and substance use disorders—that substantially interfered with or limited one or more major life activities [30]); and (d) nicotine dependence [31], *DSM-5* disorder of any psychotherapeutics use (including prescription opioids, tranquilizers, and/or sedatives), and the use of any illicit drugs excluding cannabis.

*Past-year alcohol treatment use:* The 2022 NSDUH defined substance use treatment (alcohol and/or drugs) as treatment received in the past year for the use of alcohol or drugs in an inpatient location; in an outpatient location; via telehealth; or in a prison, jail, or juvenile detention center; or the receipt of medication-assisted treatment for alcohol or opioid use. The receipt of other substance use treatment services, including a support group, a peer support specialist or recovery coach who works with a substance use treatment program or other treatment provider, services in an ED, or detoxification or withdrawal support services from a healthcare professional, was not classified as substance use treatment. In this study, we focused on alcohol-specific treatment use (which may have included other drug treatment use).

*Sociodemographic factors* were age (50–64 and 65+ years); gender; race/ethnicity; marital status (married vs. not married); education (college degree vs. no college degree); and income (<poverty line, up to 2X poverty line, >2X poverty line).

### 2.3. Analysis

We used Stata/MP 18′s svy function (College Station, TX, USA) and subpop command in all analyses to account for NSDUH’s multi-stage, stratified sampling estimates to ensure that variance estimates incorporate the full sampling design. All estimates presented in this study are weighted except sample sizes. Firstly, we calculated the prevalence, with 95% confidence intervals, of past-year alcohol use, past-month binge use, past-year AUD, and alcohol risk perception first among all individuals aged 50 and older and then among past-year alcohol users. Secondly, focusing on past-year alcohol users, we used Pearson’s χ^2^ and *t* tests to compare those with and without the perception of great risk with respect to sociodemographic characteristics, physical and mental health, other substance use, alcohol use frequency, binge use frequency, AUD, and alcohol treatment use. Thirdly, we fitted three generalized linear models (GLMs) for a Poisson distribution with a log link function to examine the associations of risk perception with (a) alcohol use frequency, (b) binge alcohol use, and (c) alcohol use disorder. GLM results are presented as incidence rate ratios (IRR) with 95% confidence intervals (CI). We used the Poisson distribution with a log link rather than logistic regression models because odds ratios exaggerate the true relative risk to some degree when the event (i.e., great risk perception) is a common (i.e., >10%) occurrence [32]. We also used a GLM to test the association between past-year alcohol treatment use and alcohol risk perception. As a preliminary diagnostic, the variance inflation factor (VIF), using a cut-off of 2.50 [33], from linear regression models was used to assess multicollinearity among covariates. VIF diagnostics indicated that multicollinearity was not a concern. Significance was set at *p <* 0.05.

## 3. Results

### 3.1. Prevalence of Alcohol Use, Binge Drinking, AUD, and Alcohol Risk Perception

Table 1 shows that of U.S. adults aged 50 and older, 61.5% (66.4% of those age 50–64 and 56.1% of those age 65+, *p* < 0.001) reported using alcohol in the preceding 12 months; 16.3% (22.0% of those age 50–64 and 10.1% of those age 65+, *p* < 0.001) reported past-month binge use; and 7.3% (10.0% of those age 50–64 and 4.3% of those age 65+, *p* < 0.001) met AUD criteria. A slightly higher proportion of the 65+ age group than the 50–64 age group reported a perceived risk of binge drinking 1–2 times a week.

Among alcohol users, 26.5% (33.3% of those age 50–64 and 17.8% of those age 65+, *p* < 0.001) reported binge use; 11.8% (15.0% of those age 50–64 and 7.6% of those age 65+, *p* < 0.001) had AUD; and 40.1% (38.8% of those age 50–64 and 41.8% of those age 65+, *p* = 0.112) perceived great risk of binge use.

Additional analysis comparing past-year alcohol users and non-users showed that a significantly lower proportion of users (40.1%) than non-users (55.0%) reported great risk perception (F[1, 50] = 134.10, *p* < 0.001). Additional analyses also showed that past-year alcohol users, regardless of their risk perception, had relatively good physical health (i.e., an average of just one chronic illness, compared to nonusers of alcohol who had an average of 1.5 [SE = 0.03] chronic illnesses; t = 4.20, *p* < 0.001), and that alcohol users and nonusers did not significantly differ on any mental illness (F[2.83, 141.42] = 2.62, *p* = 0.056).

### 3.2. Characteristics of Past-Year Alcohol Users by Risk Perception

Table 2 shows that compared to those who did not perceive a great risk of binge use, those who perceived a great risk included higher proportions of females, racial/ethnic minorities, and college graduates and lower proportions of cannabis users and those with nicotine dependence. Those who perceived a great risk also included lower proportions of near daily/daily drinkers, those with DUI, and binge and heavy drinkers, and those with AUD. However, the two risk perception groups did not significantly differ in age, marital status, income, number of chronic illness, any ED visit, any mental illness, cannabidiol use, psychotherapeutics use disorder, illicit drug use, and alcohol treatment use. Less than 2% of past-year alcohol users reported using alcohol treatment; however, additional analysis found that among those with AUD, 9.8% (15.7% and 7.7% of those with and without great risk perception, respectively, F[1, 50] = 3.81, *p* = 0.057) received alcohol treatment.

### 3.3. Associations of Risk Perception with Alcohol Use Frequency, Binge Use Frequency, and AUD

Table 3 shows that past-year near daily (IRR = 0.78, 95%CI = 0.69–0.87) or daily use (IRR = 0.60, 95%CI = 0.48–0.74) of alcohol, any past-month binge use, regardless of frequency (IRR = 0.33, 95%CI = 0.19–0.57 for 6–19 days of binge use), and mild AUD (IRR = 0.71, 95%CI = 0.54–0.94) were associated with a lower likelihood of great risk perception. Moderate and severe AUDs were not associated with the risk perception. Of the covariates, female gender, minority status, and having a college degree were associated with a higher likelihood of great risk perception, whereas DUI and cannabis use were associated with a lower likelihood. ED visits were associated with a higher likelihood of great risk perception only in Model 2 where binge use was the primary variable of interest. Nicotine dependence was associated with a lower likelihood of great risk perception only in Model 3 where AUD was the primary variable of interest.

### 3.4. Association of Alcohol Treatment Use and Risk Perception

Table 4 shows that alcohol treatment use was not associated with alcohol risk perception. The only significant correlates of treatment use were AUD (IRR = 47.5, 95%CI = 18.48–121.88 for severe AUD), mild mental illness (IRR = 2.18, 95%CI = 1.03–4.61), and any psychotherapeutic use disorder (IRR = 2.32, 95%CI = 1.06–5.09).

## 4. Discussion

The study findings show that 62% of U.S. individuals age 50 and older used alcohol in the past year. Among past-year alcohol users, 27% reported past-month binge drinking, 12% had AUD, and 40% (without any significant difference between the 50–64 and the 65+ age groups) perceived a great risk of binge drinking 1–2 times a week. The findings from multivariable models show that near daily/daily alcohol use, any frequency of binge use, and mild AUD were associated with a lower likelihood of perceiving a great risk of binge use.

The temporal order of the associations between risk perception and frequent and binge use and AUD cannot be established in cross-sectional data. Those with low risk perception may consume more alcohol. At the same time, those who engage in problematic alcohol use may rationalize their use by attributing a low risk to such consumption to have the self-image of a controlled and responsible drinker [22]. Previous studies found that low alcohol-related risk perception among older alcohol users may in part be responsible for continued harmful use [25]. Our findings regarding the negative associations between the perceived great risk of binge alcohol use and near daily/daily use, binge use and DUI also point to the possibility of older alcohol users rationalizing their drinking habits as less risky. Older adults who have had long-term drinking patterns may have difficulty changing habits and may not want to admit any related risks even if they experience negative effects from their drinking. Given the confusing array of messages regarding alcohol risks and benefits (e.g., protective health effects of moderate consumption), some older adults may not have updated and accurate knowledge about drinking guidelines and alcohol-related harms [34,35].

Our finding that mild AUD was associated with a lower likelihood of perceived great risk is concerning, as those with a low risk perception may continue their problematic alcohol use. The lack of association between binge alcohol risk perception and moderate and severe AUD may be due to the small sample size in the study. However, it is also possible that those with serious alcohol problems did not want to admit any alcohol-related risks in order to justify their drinking. Older adults with a long-term history of problematic use may also have become desensitized to alcohol-related dangers and/or had impaired judgment to perceive risks accurately. While no previous studies examined the associations between alcohol risk perception and AUD in older adults, a previous study found that older adults with cannabis use disorder were significantly more likely to endorse great risk perceptions of smoking cannabis 1–2 times a week [36]. More research is needed to examine the associations between alcohol risk perception and AUD in older adults and why binge alcohol risk perception among those with AUD may be different from cannabis smoking risk perception among those with cannabis use disorder.

Our findings also show negative associations between binge use risk perception and cannabis use and nicotine dependence among older alcohol users. Given the rapid increase in cannabis use among older adults, the co-use of cannabis along with binge drinking is not surprising [18,20]. However, it is not clear why cannabis use is negatively associated with alcohol risk perception. It is possible that older adults who co-use cannabis and alcohol may have lower risk perceptions about substance use in general and are more likely to engage in the problematic use (e.g., binge alcohol use). Another study found that tobacco use and illicit drug use disorder were associated with increased odds of binge drinking among those age 50 and older [37]. Consistent with previous study findings [18], we found that past-year alcohol users had better health than non-users. However, continued problematic alcohol use is likely to exert detrimental effects on health, especially among poly-substance users. A study of Swedish individuals age 50 and older with alcohol use problems found that the incidents of alcohol-related hospitalizations were higher, especially among those with poly-drug use problems and psychiatric problems [38].

With respect to the findings related to sociodemographic correlates, higher alcohol-related risk perceptions among females [39] are well known and likely reflect the gender differences in drinking motives and behaviors. For example, a study of disabled adults (median age of 59 years) found that the effects of depressive symptoms on problematic drinking were significantly greater for males than for females, suggesting that males may be more likely to resort to problematic alcohol use to cope with negative affective symptoms [40]. A multi-national study of alcohol consumption in older adults across ten years also found that heavy drinking, along with co-occurring medical conditions, health behaviors, and female sex, was consistently associated with a higher hazard of depressive episodes [41].

Higher risk perception regarding binge use among minoritized older adults compared to their non-Hispanic white peers may be influenced by several personal and community-related factors and situational norms. Given health disparities among minoritized older adults, awareness of their already heightened health risks might lead them to perceive binge drinking as an additional threat to their well-being. These older adults may also have witnessed more adverse effects of problematic alcohol use within their communities, such as higher rates of alcohol-related health issues, accidents, and violence. Given racial profiling at traffic stops [42], they are also likely to be keenly aware of the legal consequences of drinking and driving. An earlier study found that Black males attributed higher jail penalties from DUI conviction [43]. The cultural and social norms of older persons as respected elders in minority communities may also make them more cautious about responsible versus irresponsible behaviors. Higher risk perception among those with college degrees may be a result of a greater level of knowledge and awareness about alcohol harms among these individuals as they may have had increased access to such information and data and higher health literacy.

The present study also shows that only AUD severity and any psychotherapeutics use disorder were significantly associated with alcohol treatment use. The treatment use rate of under 10% among those with AUD was low, even though the definition of treatment is restricted to outpatient or inpatient locations or to medication-assisted treatment. That 9 out of 10 older adults with AUD did not use any alcohol treatment service is a significant problem, since behavioral health (cognitive behavioral therapy, motivational enhancement therapy, acceptance- and mindfulness-based interventions, contingency management approaches, couples and family counseling, Alcoholics Anonymous, and 12-step facilitation therapy) and FDA-approved pharmacological treatments (disulfiram, acamprosate, and naltrexone) for AUD are effective [15,44]. A systematic review showed that older adults respond well to behavioral interventions aimed at reducing alcohol consumption [44,45], although there are significant gaps in research concerning the three FDA-approved pharmacologic treatment options for AUD in older adults [46].

This study’s limitations need mentioning. Firstly, since the NSDUH is based on self-report data, the validity of respondents’ self-reported behaviors and other variables was not ascertained. The reporting of alcohol use frequency, binge use, and other substance use may have been affected by social desirability bias. Secondly, the NSDUH’s binge drinking risk perception question was not specific about different types of risk (e.g., effect on health, social/legal consequences, harm to self or others). Respondents may have different risk perceptions of different domains. Thirdly, while the overall sample size is large, the number of those with AUD was small and may have affected the results. Fourthly, as the 2022 NSDUH adopted a narrow definition of substance use treatment (e.g., exclusion of support groups, ED treatment), this study’s findings of treatment use rate should not be compared to the rates in the studies that adopted a wider definition of treatment. Fifthly, only correlation, not causation, can be inferred from cross-sectional data.

## 5. Conclusions

About 40% of U.S. individuals age 50 and older who used alcohol in the past year perceived a great risk of binge drinking use 1–2 times a week. However, the likelihood of the great risk perception was lower among frequent alcohol users, binge users, and those with mild AUD. Risk perception was not significantly associated with alcohol treatment use. Although AUD was significantly associated with alcohol treatment use, less than one in ten individuals with AUD received treatment in this age group.

Lower alcohol risk perception among potentially problematic alcohol users is concerning given the fact that the negative health effects of alcohol are more deleterious in older than younger age groups. The study’s findings point to the importance of the following educational efforts, interventions, and policies. Firstly, there is a need for educational materials that include the specific health concerns and interests of older adults and are distributed on a wide variety of platforms (e.g., media channels, mobile apps, online resources) to increase alcohol risk perceptions. Studies have shown that providing information and educational materials on the consequences of alcohol consumption were effective in reducing problematic alcohol use across age groups [47,48]. Secondly, healthcare providers need to screen and provide brief interventions and personalized counseling to older-adult problem drinkers addressing the individual’s specific circumstances and health status. Thirdly, older adults with AUD need to utilize alcohol treatment. Availability, accessibility, and affordability of treatment programs for late-middle-aged and older adults need to be improved. Fourthly, more research is needed to examine alcohol risk perception and its association with problematic alcohol consumption and treatment use.

## Figures and Tables

**Table 1 ijerph-21-01081-t001:** Prevalence of alcohol use, binge drinking, alcohol use disorder, and binge alcohol use perception in U.S. individuals age 50 and older (% with 95% confidence intervals).

	Aged 50 and Older	Aged 50–64	Aged 65 and Older	*p* ^a^
**Of all older adults**				
Sample N	10,560	5369	5191	
Population size	118,705,224	61,928,903	56,776,322	
Past-year alcohol use	61.5 (59.9–63.1)	66.4 (64.3–68.5)	56.1 (53.6–58.6)	<0.001
Past-month binge drinking	16.3 (15.2–17.4)	22.0 (20.4–23.7)	10.1 (8.6–11.6)	<0.001
Alcohol use disorder	7.3 (6.5–8.1)	10.0 (8.6–11.5)	4.3 (3.5–5.2)	<0.001
Perceived risk of binge alcohol use 1–2 times a week				0.015
No risk	2.6 (2.1–3.3)	2.7 (2.0–3.7)	2.4 (1.9–3.2)	
Slight risk	13.8 (12.8–14.8)	15.4 (13.8–17.2)	12.0 (10.8–13.3)	
Moderate risk	36.7 (35.1–38.2)	36.5 (34.5–38.5)	36.8 (34.7–39.1)	
Great risk	45.8 (44.0–47.7)	44.3 (41.9–46.8)	47.5 (45.1–49.8)	
Missing	1.1 (0.9–1.5)	1.0 (0.7–1.4)	1.3 (0.8–2.0)	
**Of past year alcohol users**				
Sample N	6693	3631	3062	
Population size	72,994,534			
Past-month binge drinking	26.5 (24.8–28.2)	33.3 (30.9–35.5)	17.8 (16.5–20.5)	<0.001
Alcohol use disorder	11.8 (10.5–13.2)	15.0 (13.0–17.3)	7.6 (6.3–9.2)	<0.001
Perceived risk of binge alcohol use 1–2 times a week				0.198
No risk	2.8 (2.1–3.7)	2.7 (1.8–3.9)	3.0 (2.1–4.3)	
Slight risk	16.8 (15.5–18.2)	18.2 (16.3–20.3)	15.0 (13.1–17.2)	
Moderate risk	39.5 (37.3–41.7)	39.6 (36.9–42.5)	39.3 (36.7–41.9)	
Great risk	40.1 (38.0–41.8)	38.8 (36.1–41.6)	41.8 (39.0–44.7)	
Missing	1.1 (0.9–1.5)	1.0 (0.7–1.4)	1.3 (0.8–2.0)	

^a^ Probability values denote differences between the two age groups and were calculated based on Pearson’s χ^2^ tests. Note: Prevalence data were calculated based on the 2022 National Survey on Health and Drug Use public use data.

**Table 2 ijerph-21-01081-t002:** Characteristics of past-year alcohol users by perceived great risk of binge alcohol use 1–2 times a week.

N (%)	No Great Risk Perception	Great Risk Perception	*p*
	3919 (59.9%)	2774 (40.1%)	
Age group (%)			0.112
50–64 years	57.6	54.5	
65+ years	42.4	45.5	
Gender (%)			<0.001
Male	55.0	42.8	
Female	45.0	57.2	
Race/ethnicity (%)			<0.001
Non-Hispanic white	78.1	69.6	
Non-Hispanic Black	7.4	10.6	
Hispanic	9.5	13.4	
All other	5.0	6.5	
Married (%)	63.3	61.1	0.233
College degree (%)	35.5	41.5	0.005
Income (%)			0.502
Below poverty line	6.8	7.8	
Up to 2X poverty line	14.7	13.5	
More than 2X poverty line	78.5	78.7	
No. of chronic illnesses (M, SE)	1.0 (0.02)	1.02 (0.03)	0.292
Any emergency department visit (%)	24.6	23.6	0.515
Past-year any mental illness (%)			0.503
None	86.6	87.9	
Mild	6.7	6.2	
Moderate	3.8	3.7	
Severe	2.9	2.1	
Cannabis use (%)	19.9	13.0	<0.001
Cannabidiol (CBD) use (%)	20.7	19.2	0.375
Nicotine dependence (%)	9.2	5.5	0.004
Disorder of any psychotherapeutics ^a^ use (%)	2.5	2.5	0.937
Illicit drug (other than cannabis) use (%)	6.3	5.2	0.290
Alcohol use frequency (%)			<0.001
1–11 days	19.3	28.0	
12–49 days	21.6	25.6	
50–99 days	12.5	13.2	
100–299 days	34.2	26.9	
300–365 days	12.3	6.3	
Driving under the influence of alcohol (%)	10.1	4.0	<0.001
Alcohol binge use (%)	34.3	14.9	<0.001
None	65.7	85.1	
1–2 days	16.8	10.1	
3–5 days	9.5	2.7	
6–19 days	5.7	1.3	
20–30 days	2.3	0.8	
Alcohol heavy use (%)	10.3	2.4	<0.001
Alcohol use disorder (%)	14.4	7.9	<0.001
None	85.6	92.1	
Mild	9.5	5.0	
Moderate	2.8	1.1	
Severe	2.1	1.8	
Alcohol treatment use (%)	1.5	1.6	0.921
Any substance (including alcohol) treatment use (%)	3.8	3.8	0.981
Any substance use support group participation (%)	1.5	1.1	0.537

^a^ Prescription opioids, tranquilizers, or sedatives. Note: Probability values were calculated based on Pearson’s χ^2^ and t tests.

**Table 3 ijerph-21-01081-t003:** Associations of binge alcohol use risk perception with alcohol use frequency, binge use frequency, and alcohol use disorder among past-year alcohol users. Results from generalized linear modeling.

	Perceived Great Risk of Binge Use vs. No Perceived Great Risk Perception		
	Model 1	Model 2	Model 3
	IRR (95% CI)	IRR (95% CI)	IRR (95% CI)
Alcohol use frequency: vs. 1–11 days			
12–49 days	0.92 (0.80–1.05)		
50–99 days	0.89 (0.76–1.03)		
100–299 days	0.78 (0.69–0.87) ***		
300–365 days	0.60 (0.48–0.74) ***		
Binge drinking days: vs. none			
1–2 days		0.64 (0.54–0.76) ***	
3–5 days		0.39 (0.29–0.52) ***	
6–19 days		0.33 (0.19–0.57) ***	
20–30 days		0.48 (0.25–0.93) ***	
Alcohol use disorder: vs. none			
Mild			0.71 (0.54–0.94) *
Moderate			0.64 (0.38–1.07)
Severe			1.29 (0.88–1.89)
65+ years vs. 50–64 years	1.07 (0.97–1.17)	0.99 (0.91–1.08)	1.04 (0.95–1.14)
Female vs. Male	1.27 (1.16–1.38) ***	1.25 (1.15–1.36) ***	1.30 (1.19–1.42) ***
Race/ethnicity: vs. non-Hispanic white			
Non-Hispanic Black	1.32 (1.14–1.52) ***	1.37 (1.19–1.58) ***	1.34 (1.17–1.54) ***
Hispanic	1.28 (1.09–1.51) **	1.32 (1.13–1.54) **	1.33 (1.13–1.55) **
Other	1.19 (1.02–1.39) *	1.22 (1.03–1.42) *	1.23 (1.06–1.44) **
Married/partnered vs. not married	0.95 (0.85–1.07)	0.94 (0.84–1.05)	0.94 (0.84–1.05)
College degree vs. no degree	1.21 (1.09–1.34) ***	1.13 (1.03–1.25) *	1.18 (1.07–1.31) **
Income: vs. below poverty line			
Up to 2X poverty line	0.86 (0.70–1.07)	0.85 (0.68–1.05)	0.87 (0.70–1.07)
More than 2X poverty line	0.94 (0.76–1.16)	0.90 (0.73–1.10)	0.94 (0.76–1.15)
No. of chronic medical condition	1.00 (0.97–1.04)	1.00 (0.96–1.04)	1.02 (0.98–1.06)
ED visit: vs. no ED visit	1.03 (0.92–1.15)	1.04 (1.03–1.25) ***	1.02 (0.92–1.14)
Past-year mental illness: vs. none			
Mild	0.91 (0.76–1.09)	0.85 (0.68–1.05)	0.94 (0.79–1.12)
Moderate/severe	0.92 (0.72–1.17)	0.93 (0.73–1.19)	0.94 (0.73–1.22)
Alcohol DUI vs. no DUI	0.61 (0.49–0.75) ***	0.67 (0.53–0.83) **	0.57 (0.46–0.71) ***
Cannabis use vs. non-use	0.80 (0.68–0.93) **	0.81 (0.70–0.94) **	0.79 (0.68–0.92) **
CBD use vs. non-use	1.06 (0.93–1.20)	1.07 (0.95–1.21)	1.06 (0.93–1.20)
Nicotine dependence vs. none	0.78 (0.60–1.02)	0.83 (0.65–1.07)	0.76 (0.58–0.99) *
Disorder of any psychotherapeutics use vs. no disorder	1.07 (0.76–1.51)	1.11 (0.79–1.57)	1.12 (0.80–1.57)
Illicit drug use excluding cannabis vs. non-use	1.03 (0.78–1.36)	1.05 (0.80–1.37)	1.01 (0.77–1.33)
	N= 6615	N = 6615	N = 6615

* *p* < 0.05; ** *p* < 0.01; *** *p* < 0.001.

**Table 4 ijerph-21-01081-t004:** Associations of past-year alcohol treatment use with binge alcohol use risk perception among past-year alcohol users.

	Treatment Use vs. Non-Use
	IRR (95% CI)
Great alcohol risk perception vs. no great risk perception	1.44 (0.77–2.71)
Alcohol use disorder: vs. none	
Mild	5.09 (1.72–15.06) **
Moderate	22.51 (7.87–64.37) ***
Severe	47.5 (18.48–121.88) ***
65+ years vs. 50–64 years	1.70 (0.88–3.28)
Female vs. Male	0.65 (0.35–1.21)
Race/ethnicity: vs. non-Hispanic white	
Non-Hispanic Black	0.85 (0.30–2.29)
Hispanic	1.55 (0.53–4.52)
Other	1.80 (0.65–4.95)
Married/partnered vs. not married	0.68 (0.41–1.11)
College degree vs. no degree	0.82 (0.46–1.44)
Income: vs. below poverty line	
Up to 2X poverty line	0.69 (0.30–1.60)
More than 2X poverty line	0.71 (0.29–1.76)
No. of chronic medical condition	0.86 (0.67–1.11)
ED visit: vs. no ED visit	1.20 (0.66–2.19)
Past-year mental illness: vs. none	
Mild	2.18 (1.03–4.61) *
Moderate/severe	1.78 (0.65–4.86)
Alcohol DUI vs. no DUI	0.85 (0.47–1.54)
Cannabis use vs. non-use	1.12 (0.67–1.88)
CBD use vs. non-use	1.50 (0.85–2.65)
Nicotine dependence vs. none	1.67 (0.92–3.02)
Disorder of any psychotherapeutics use vs. no disorder	2.32 (1.06–5.09) *
Illicit drug use excluding cannabis vs. non-use	0.89 (0.36–2.21)
	N= 6615

* *p* < 0.05; ** *p* < 0.01; *** *p* < 0.001.

## Data Availability

This study is based on de-identified public-domain data (The National Survey on Drug Use and Health).

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
