# Peer review of "Perceived Risk of Binge Drinking among Older Alcohol Users: Associations with Alcohol Use Frequency, Binge Drinking, Alcohol Use Disorder, and Alcohol Treatment Use"

_ijerph, 2024, doi:10.3390/ijerph21081081_

Round 1
Reviewer 1 Report
Comments and Suggestions for Authors
Perceived Risk of Binge Drinking among Older Alcohol Users: Associations with Alcohol Use Frequency, Binge Drinking, Alcohol Use Disorder, and Alcohol Use Treatment Receipt (Choi, et al.)
Thank you for allowing me to review this interesting and well written article. It's greatest strengths are that the methods are well done, and it does add to the literature. I have made some specific comments below. Most of my comments are minor and involve small writing and conceptual issues.
Up front, I would like to say that this article is expansive and might even be better as two separate articles. I leave that to the author(s) and editors.
1. The first sentence is questionable and needs to be clarified. Alcohol is not the most harmful to all people, and this sounds as if it is. Even the supporting articles are clear on this. Second sentence, I would take out “The” at the beginning because it looks like you are continuing with the previous line of thought that is not entirely accurate. Just say, “Negative health effects…”
2. As I was reading the paragraph in lines 73-84, it occurred to me that you might even add that many older adults also recognize some positives from drinking such as maintaining social connections, etc. I am not sure which of the articles you cited mentioned that, I know some did, though.
3. As I was reading the paragraph in lines 270-283, I thought that maybe the differences between perceived risks associated with cannabis vs alcohol are not at all surprising. These are people who have grown up with much stigma associated with cannabis vs. alcohol. That is mentioned in the next to last sentence. I would like that to maybe be expanded. Also, is it possible that those with more serious issues do not report risk for other reasons than small sample size? For example, maybe they drink more because they don’t see it as risky. It could also be rationalization (which you mention earlier), too, in fact, meaning the causation could go both ways.
4. Paragraph beginning with line 299…Maybe it is the other way around…maybe more alcohol use leads to more depressive symptomology? I didn’t check the references, but that does sound reasonable to me.
Author Response
Thank you for allowing me to review this interesting and well written article. It's greatest strengths are that the methods are well done, and it does add to the literature. I have made some specific comments below. Most of my comments are minor and involve small writing and conceptual issues.
Up front, I would like to say that this article is expansive and might even be better as two separate articles. I leave that to the author(s) and editors.
RESPONSE: Thank you very much for your time reviewing our paper. We are grateful to you for very helpful comments and have incorporated/addressed all of them in the revision. The text word count is less than 4,300 words, so we believe it is within the range of the journal guidelines and would like to keep it as a single paper.
- The first sentence is questionable and needs to be clarified. Alcohol is not the most harmful to all people, and this sounds as if it is. Even the supporting articles are clear on this. Second sentence, I would take out “The” at the beginning because it looks like you are continuing with the previous line of thought that is not entirely accurate. Just say, “Negative health effects…”
RESPONSE: Thank you for these comments. Beyond and above the four papers that we cited, we have done more research on the comparative harms of substances and confirmed that alcohol is the most dangerous substance. According to one source, “alcohol has shortened the lifespan of those 88,000 human beings by 30 years. All other drugs combined only cause 30,000 deaths a year” (https://www.addictioncenter.com/community/why-alcohol-is-the-deadliest-drug/). "The negative health effects" were changed to "Negative health effects."
- As I was reading the paragraph in lines 73-84, it occurred to me that you might even add that many older adults also recognize some positives from drinking such as maintaining social connections, etc. I am not sure which of the articles you cited mentioned that, I know some did, though.
RESPONSE: Done. In the fourth paragraph in the Introduction section, we cited research that shows that older adults appreciate alcohol use in sustaining their social activities and connections.
- As I was reading the paragraph in lines 270-283, I thought that maybe the differences between perceived risks associated with cannabis vs alcohol are not at all surprising. These are people who have grown up with much stigma associated with cannabis vs. alcohol. That is mentioned in the next to last sentence. I would like that to maybe be expanded. Also, is it possible that those with more serious issues do not report risk for other reasons than small sample size? For example, maybe they drink more because they don’t see it as risky. It could also be rationalization (which you mention earlier), too, in fact, meaning the causation could go both ways.
RESPONSE: At the end of the second paragraph and the third paragraph of the Discussion section, we added more research-based potential reasons for low risk perception among problem drinkers. Although we reported the differences in risk perceptions related to cannabis smoking and binge alcohol use and their associations with cannabis use disorder and alcohol use disorder, we did not find sufficient research base to speculate the reasons behind those differences. Thus, our revisions are as follows:
“Given the confusing array of messages regarding alcohol risks and benefits (e.g., protective health effects of moderate consumption), some older adults may not have updated and accurate knowledge about the drinking guidelines and alcohol-related harms [34,35]
Our finding that mild AUD was associated with a lower likelihood of perceived great risk is concerning, as those with a low risk perception may continue their problematic alcohol use. The lack of association between binge alcohol risk perception and moderate and severe AUD may be due to the small sample size in the study. However, it is also possible that those with serious alcohol problems did not want to admit any alcohol-related risks in order to justify their drinking. Older adults with a long-term history of problematic use may also have become desensitized to alcohol-related dangers and/or had impaired judgment to perceive risks accurately. While no previous studies examined the associations between alcohol risk perception and AUD in older adults, a previous study found that older adults with cannabis use disorder were significantly more likely to endorse great risk perceptions of smoking cannabis 1-2 times a week [36]. More research is needed to examine the associations between alcohol risk perception and AUD in older adults and why binge alcohol risk perception among those with AUD may be different from cannabis smoking risk perception among those with cannabis use disorder.”
- Paragraph beginning with line 299…Maybe it is the other way around…maybe more alcohol use leads to more depressive symptomology? I didn’t check the references, but that does sound reasonable to me.
RESPONSE: Indeed! We added the following sentence: “A multi-national study of alcohol consumption in older adults across ten years also found that heavy drinking, along with co-occurring medical conditions, health behaviors, and female sex, was consistently associated with a higher hazard of depressive episodes [41].”
Reviewer 2 Report
Comments and Suggestions for Authors
This manuscript uses 2022 NSDUH data to identify and explore associations between perceptions of risk from alcohol use among adults 50 and older. This manuscript is extremely important, as it expands the field’s understanding of older adult alcohol use at a critical time. I have a few minor comments that I hope will help the authors strengthen this already strong manuscript.
Results
1. I found the tables very difficult to follow—partly for the centering rather than left justified format that is usually used when presenting results—it’s hard to tell what are subcategories and what are stand alone variables.
2. Table 3 and section 3.3. I found this difficult to follow—especially when the authors use “the independent variable” to describe the three distinct drinking variables. I presume the authors mean “focal predictor” or “predictor of interest”? Using IV makes it seem like this is experimental research. Given that there is a host of covariates that are discussed, it seems some of them are considered predictors as well. It would help if the table was again left justified and then Models 1, 2 and 3 were better defined in the narrative.
Discussion
3. Line 309 of the discussion. Discussion of minoritized adults. I would strongly caution the authors in their discussion of minoritized adults. These statements conflate race alone with the experiences of minority races and ethnicities who also experience poverty, environmental unsound living conditions, food deserts, etc. These are not the same thing. The Caetano article cited to support this assertion or possible explanation for the results is about 30 years old, when there was less awareness that these assertions stem from stereotypes. An alternative reason for race alone as perceiving more risk is structural racism. In such a system, individuals are aware that risks for use and any subsequent interaction with police and/or treatment programs or other social services can have dire or deadly consequences for these families. Furthermore, it’s possible that race and ethnic minority adults have more spiritual and religious connections that may influence attitudes towards the risks of alcohol.
4. Line 347. “Barely”—I would use less than one in ten.
Author Response
This manuscript uses 2022 NSDUH data to identify and explore associations between perceptions of risk from alcohol use among adults 50 and older. This manuscript is extremely important, as it expands the field’s understanding of older adult alcohol use at a critical time. I have a few minor comments that I hope will help the authors strengthen this already strong manuscript.
RESPONSE: Thank you so much for endorsing our study and your very helpful comments. We have incorporated them all in the revised version.
Results
- I found the tables very difficult to follow—partly for the centering rather than left justified format that is usually used when presenting results—it’s hard to tell what are subcategories and what are stand alone variables.
RESPONSE: We see that the journal typesetter used centering for the cells. This needs to be corrected and the format should be the same as in our manuscript. We will ask the journal typesetters to correct the formatting issues.
- Table 3 and section 3.3. I found this difficult to follow—especially when the authors use “the independent variable” to describe the three distinct drinking variables. I presume the authors mean “focal predictor” or “predictor of interest”? Using IV makes it seem like this is experimental research. Given that there is a host of covariates that are discussed, it seems some of them are considered predictors as well. It would help if the table was again left justified and then Models 1, 2 and 3 were better defined in the narrative.
RESPONSE: Thank you for pointing these out. We replaced the “independent variable” with the “primary variable of interest.”
Discussion
- Line 309 of the discussion. Discussion of minoritized adults. I would strongly caution the authors in their discussion of minoritized adults. These statements conflate race alone with the experiences of minority races and ethnicities who alsoexperience poverty, environmental unsound living conditions, food deserts, etc. These are not the same thing. The Caetano article cited to support this assertion or possible explanation for the results is about 30 years old, when there was less awareness that these assertions stem from stereotypes. An alternative reason for race alone as perceiving more risk is structural racism. In such a system, individuals are aware that risks for use and any subsequent interaction with police and/or treatment programs or other social services can have dire or deadly consequences for these families. Furthermore, it’s possible that race and ethnic minority adults have more spiritual and religious connections that may influence attitudes towards the risks of alcohol.
RESPONSE: We really appreciate these comments. We deleted the Caetano article, and added the following sentence in the paragraph: “Given the racial profiling in traffic stops [42], they are also likely to be keenly aware of the legal consequences of drinking and driving. An earlier study found that Black males attributed higher jail penalties to DUI conviction [43].”
- Line 347. “Barely”—I would use less than one in ten.
RESPONSE: Done.
Reviewer 3 Report
Comments and Suggestions for Authors
Dear authors, I completed the review of your paper, it is very interesting, well presented i have few comments to be corrected
Line 33 to 34 please rephrase the sentence is diffuclt to be understood
Line 48 first apparition of DSM please define it
Line 98 please define SAMHSA
Line 108 please verifiy if it is age or aged
Line 152 define also COPD
Line 165 define ED
Please rephrase lines 200 and 201
Plase correct the tilte of table 1
In the first line of table 2 please add %
Correct the title of table 3
Comments on the Quality of English Languagesome sentences needs to be rephrased for more clarity
Author Response
Dear authors, I completed the review of your paper, it is very interesting, well presented i have few comments to be corrected
RESPONSE: Thank you so much for your review and pointing out the needs for revision. We have corrected them all.
Line 33 to 34 please rephrase the sentence is difficult to be understood
RESPONSE: We shortened the sentence to make it to flow better.
Line 48 first appearance of DSM please define it
RESPONSE: Done.
Line 98 please define SAMHSA
RESPONSE: Done.
Line 108 please verify if it is age or aged
RESPONSE: We prefer to use “age” to “aged,” as “age” appears to be used more frequently in recent years. Some people have strong preference for it (as aged may have negative connotations especially when used with older age groups).
Line 152 define also COPD
RESPONSE: Done.
Line 165 define ED
RESPONSE: Done.
Please rephrase lines 200 and 201
RESPONSE: We shortened the sentence to make it to flow better
Please correct the title of table 1.
RESPONSE: Done.
In the first line of table 2 please add %.
RESPONSE: Done.
Correct the title of table 3.
RESPONSE: Done.